# ProtAttn-QuadNet: An attention-based deep learning framework for protein–protein interaction prediction using ProtBERT embeddings

Md. Shahidul Islam◉*, Md. Muhtasim Rahman Mim◉, Md. Raihan Kabir

Department of Computer Science and Engineering, University of Asia Pacific, Dhaka, Bangladesh

* shahid@uap-bd.edu

## Abstract

Protein–protein interactions (PPIs) form the backbone of most cellular processes, governing signal transduction, gene regulation, and metabolic control. However, experimental approaches to identifying PPIs remain expensive, laborious, and often incomplete. Recent advances in protein language models (PLMs) have transformed sequence-based PPI prediction by enabling deep contextual encoding of biochemical and structural information directly from amino acid sequences. Building upon this progress, we present ProtAttn-QuadNet, an attention-based deep learning framework that leverages ProtBERT embeddings to model reciprocal dependencies between protein pairs. The proposed model employs a quad-stream attention mechanism that integrates individual protein features, synergistic interactions, and complementary differences through multi-level self- and cross-attention layers. This architecture enables the discovery of fine-grained relational patterns while ensuring balanced bidirectional modeling of interacting proteins. Evaluated on the independent test set of a large-scale dataset from UniProt, ProtAttn-QuadNet achieves 97.16% accuracy (AUC-ROC 99.00%) on balanced data and 99.19% accuracy (AUC-ROC 99.76%) on oversampled datasets, surpassing several recent state-of-the-art PPI prediction methods. Statistical validation using the Chi-square and Wilcoxon signed-rank tests confirms the model's predictive significance and reliability. ProtAttn-QuadNet offers a powerful computational framework for large-scale PPI prediction.

## Introduction

Protein–protein interactions (PPIs) are fundamental to almost all cellular processes, including signal transduction, gene expression regulation, metabolic control, and immune responses [1–3]. Understanding the complex network of PPIs provides valuable insights into cellular functions and disease mechanisms [4,5]. Although numerous experimental techniques, such as yeast two-hybrid screening,

**Data availability statement:** The primary data are available from UniProt (https://www.uniprot.org/uniprotkb?query=reviewed:true). All reviewed (Swiss-Prot) entries from UniProtKB were used in this study, comprising 573,661 protein sequences. The processed data and code supporting this study are publicly available on Figshare and can be accessed through the following link: https://doi.org/10.6084/m9.figshare.30637145.

**Funding:** The author(s) received no specific funding for this work.

**Competing interests:** The authors have declared that no competing interests exist.

co-immunoprecipitation, and affinity purification coupled with mass spectrometry, have been developed to detect PPIs, these methods remain time-consuming, costly, and often limited in coverage [6]. Consequently, computational prediction methods have become indispensable for large-scale PPI analysis.

Early computational approaches primarily relied on handcrafted sequence features, including amino acid composition, evolutionary profiles, and physicochemical descriptors. Classical machine learning algorithms such as Support Vector Machines (SVM), Random Forests (RF), and Bayesian classifiers were employed to classify interacting protein pairs based on these features [7–12]. While these models demonstrated moderate success, their dependence on manually engineered descriptors and incomplete structural data limited their generalization capabilities, particularly across species and diverse protein families [11,13–15].

The increasing availability of large-scale protein sequence databases has encouraged sequence-based prediction methods that rely less on structural information. Deep learning has substantially advanced this field by enabling hierarchical feature extraction and representation learning. DeepPPI [16] used a fully connected neural network to model complex non-linear relationships between protein features, whereas DPPI [17] applied a Siamese-like convolutional architecture to learn symmetric relationships between interacting proteins. Similarly, PIPR [18] introduced a residual recurrent convolutional neural network (RCNN) to capture both local motifs and long-range dependencies, while Wu et al. proposed DL-PPI [19], a graph neural network–based model that integrates multi-scale features and attention mechanisms to enhance relational reasoning among proteins. These architectures collectively improved predictive performance but often struggled with interpretability, data imbalance, and computational efficiency.

Recent advances in transformer architectures and PLMs have transformed sequence-based PPI prediction by learning contextualized residue representations through self-attention mechanisms. Pretrained models such as ProtTrans [5], Prot-BERT [20], and ESM-2 [21] encode rich biochemical and evolutionary information from massive unlabeled protein corpora, effectively capturing secondary and tertiary structure tendencies directly from primary sequences. Several recent studies have leveraged these embeddings for PPI prediction using hybrid deep architectures. For example, xCAPT5 [22] integrated ProtTrans embeddings with a multi-kernel convolutional network to capture local and global dependencies, while TUnA [23] incorporated uncertainty modeling within a transformer framework to improve robustness. PPI-Graphomer [24] combined pretrained language models with graph transformers to integrate sequence and structural representations, achieving high performance across benchmark datasets.

Despite these advances, existing frameworks often treat protein pairs asymmetrically and fail to explicitly model the reciprocal dependencies inherent in protein–protein binding. Moreover, many attention-based methods focus on single-sequence encoding and overlook bidirectional relationships critical to interaction dynamics.

To address these limitations, this study proposes ProtAttn-QuadNet, an attention-based deep learning framework that leverages ProtBERT embeddings to

model mutual interactions between protein pairs. By incorporating quadratic attention mechanisms and a reciprocal representation module, ProtAttn-QuadNet identifies key sequence regions contributing to binding affinity and ensures balanced interaction assessment. The proposed model was evaluated on dataset collected from UniProt [25] and demonstrated high predictive accuracy and robustness across species. Overall, ProtAttn-QuadNet offers a reliable computational framework for large-scale PPI prediction and contributes to a deeper understanding of cellular interaction networks.

## Materials and methods

### Dataset

A total of 573,661 protein entries were collected from the UniProtKB database [25]. Proteins were divided into interacting and non-interacting groups. Duplicate interaction pairs were removed; if both A=B and B=A were present, only one was kept. Each protein pair was labeled 1 for interaction (positive) or 0 for non-interaction (negative). To balance the dataset, two approaches were applied: (a) selecting an equal number of positive and negative samples, resulting in 249,814 protein pairs (124907 positive, 124907 negative) and 157,839 unique proteins, and (b) oversampling the positive class to include all proteins, resulting in 1,082,662 protein pairs (541331 positive, 541331 negative) and 573,661 unique proteins.

A separate dataset of 573,661 protein sequences was used for sequence embedding. Each protein sequence was converted into a 1024-dimensional numerical vector using ProtBERT embeddings [20], which were then used as model input.

### Data preprocessing

In this study, robust scaling was used to normalize protein embedding vectors. The transformation is defined as:

$$X_{\text{scaled}} = \frac{X - \text{median}(X)}{\text{IQR}(X)}$$

(1)

where $X \in \mathbb{R}^{N \times D}$ is the protein embedding matrix with $N$ proteins and $D = 1024$ dimensions, and IQR is the interquartile range $(Q_3 - Q_1)$.

Biological data such as protein embeddings often exhibit non-Gaussian distributions, contain outliers, and represent diverse protein families, making standard normalization approaches less effective. Protein embeddings derived from neural language models or structural encoders may show heavy-tailed distributions, with extreme values arising from rare proteins, unusual structural motifs, membrane domains, or intrinsically disordered regions. Furthermore, since these embeddings are pre-computed by external models and generated outside our training pipeline, their statistical properties are not guaranteed to follow a normal distribution. Under such conditions, StandardScaler, which relies on the mean and standard deviation, can be strongly influenced by extreme values. In contrast, MinMaxScaler, which depends on the minimum and maximum values, is also sensitive to outliers that may compress the majority of the data into a narrow range. In contrast, RobustScaler centers the data using the median and scales it by the interquartile range (IQR), both of which are less affected by extreme observations. This makes robust scaling more suitable for stabilizing embedding feature distributions and improving the reliability of downstream model training.

### Dataset splitting

A stratified splitting strategy was employed to ensure balanced class distribution across training, validation, and test sets while maintaining statistical rigor for model evaluation. To achieve this, a two-stage stratified sampling approach was adopted instead of a single-stage three-way split. Single-stage splitting can introduce subtle class imbalances due to rounding effects when dividing samples into three partitions simultaneously, whereas two-stage splitting manages one binary decision at a time (train+validation vs. test, then train vs. validation), ensuring precise stratification.

In the first stage, a fully isolated test set (12%) was created and kept untouched during model development, hyper-parameter tuning, and validation to prevent data leakage. The second stage applied a test size of 0.225 to the remaining 88% of data, yielding a validation set that accounted for exactly 20% of the total dataset ($0.225 \times 0.88 \approx 0.20$). This method preserved nested stratification, maintaining the original class distributions at both stages—something not guaranteed by a single three-way split.

Formally, given a dataset $D$ with binary interaction labels $Y = y_1, y_2, \ldots, y_n$, where $y_i \in 0, 1$, stratified splitting was performed as described above to preserve the proportions of positive and negative interactions across all partitions.

Stage 1: Training+Validation vs Test Split

$$D = D_{\text{train\_val}} \cup D_{\text{test}}$$

Stage 2: Training vs Validation Split

$$D_{\text{train\_val}} = D_{\text{train}} \cup D_{\text{val}}$$

Thus, the final balanced dataset contains 169,873 training samples, 49,963 validation samples, and 29,978 test samples, while the oversampled dataset contains 736,210 training samples, 216,532 validation samples, and 129,920 test samples.

## Feature engineering

Our objective is to enhance interaction-aware representations that capture diverse functional dependencies in protein–protein interactions (PPIs). We introduce two key feature types, Element-wise Product and Absolute Difference, which collectively capture complementary interaction dynamics.

- **Element-wise Product** focuses on synergistic relationships by emphasizing dimensions where both proteins exhibit strong activations simultaneously. When both embeddings express high values in the same dimension, their product yields a large value, reflecting cooperative or co-regulatory behavior.

For protein embeddings $\mathbf{x}_1, \mathbf{x}_2 \in \mathbb{R}^d$, we define the interaction feature as:

$$\mathbf{x}_{int} = \mathbf{x}_1 \odot \mathbf{x}_2 = [x_{1,1} \cdot x_{2,1}, x_{1,2} \cdot x_{2,2}, \ldots, x_{1,d} \cdot x_{2,d}] \tag{2}$$

where, $\mathbf{x}_{int}$ captures *synergistic activation*, representing cooperative dimensions where both proteins contribute strongly.

- **Absolute Difference** on the other hand, models complementary relationships by quantifying divergence between embeddings. Large values indicate distinct functional behaviors or differing biological characteristics between the two proteins. We define a difference feature as:

$$\mathbf{x}_{diff} = |\mathbf{x}_1 - \mathbf{x}_2| = [|x_{1,1} - x_{2,1}|, |x_{1,2} - x_{2,2}|, \ldots, |x_{1,d} - x_{2,d}|] \tag{3}$$

where, $\mathbf{x}_{diff}$ captures *complementary activation*, representing the magnitude of contrast across feature dimensions.

- **Gaussian Noise Injection for Data Augmentation** To enhance the robustness and generalization of the model, we introduce Gaussian noise to the protein embeddings generated by ProtBERT. Since these embeddings capture rich sequence-level features, small perturbations simulate natural variation in protein representations and prevent the model from overfitting, and encourage the model to learn generalizable patterns of interactions rather than memorizing

specific embeddings. Formally, Gaussian noise is added independently to both protein embeddings in each interaction pair:

$$\mathbf{X}_1^{aug} = \mathbf{X}_1 + \varepsilon_1, \quad \mathbf{X}_2^{aug} = \mathbf{X}_2 + \varepsilon_2 \tag{4}$$

where $\mathbf{X}_1, \mathbf{X}_2 \in \mathbb{R}^{1024}$ are the original scaled embeddings, $\varepsilon_1, \varepsilon_2 \sim \mathcal{N}(0, \sigma^2 I)$ are independent Gaussian noise vectors, $\sigma = 0.02$, and $I \in \mathbb{R}^{1024 \times 1024}$ is the identity matrix.

## Model architecture

A multi-stream attention model is designed to predict protein–protein interactions while also estimating interaction uncertainty, binding strength, and interaction type. The model takes four different types of features as input and processes them through parallel attention streams and cross-attention layers to capture both individual protein properties and their relationships. The overall architecture of the proposed model is illustrated in Fig 1.

## Advanced attention block

The Advanced Attention Block is designed as the core computational unit of the architecture, which processes each feature stream through a sophisticated attention mechanism. Given an input $\mathbf{x} \in \mathbb{R}^{\text{batch\_size} \times \text{input\_dim}}$, it is first projected into a higher-dimensional hidden space as follows:

$$\mathbf{h} = \text{Linear}(\mathbf{x}) \in \mathbb{R}^{\text{batch\_size} \times \text{hidden\_dim}} \tag{5}$$

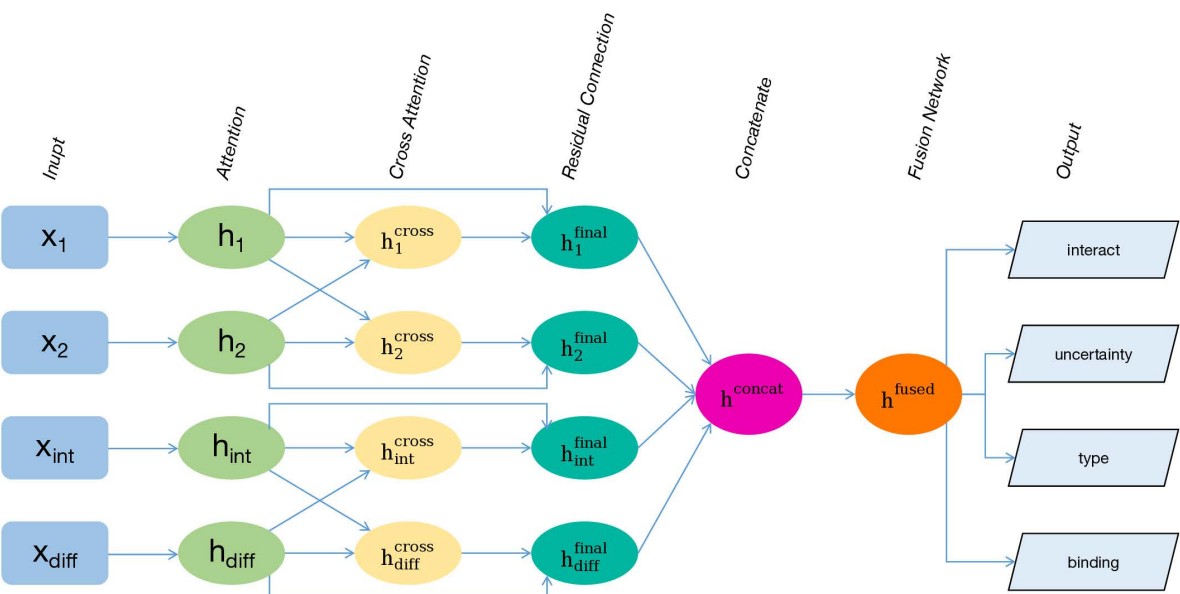

**Fig 1. Architecture of the ProtAttn-QuadNet framework.** Four distinct protein feature representations are processed through parallel attention streams and cross-attention layers to predict protein–protein interactions and simultaneously estimate interaction uncertainty, binding strength, and interaction type.

The projected features are normalized using layer normalization:

$$\mathbf{h}_{norm} = \text{LayerNorm}(\mathbf{h})$$

Subsequently, multi-head self-attention is applied:

$$\mathbf{Q} = \mathbf{K} = \mathbf{V} = \mathbf{h}_{norm}$$

$$\mathbf{A} = \text{MultiHeadAttention}(\mathbf{Q}, \mathbf{K}, \mathbf{V}, \text{num\_heads} = 8)$$

$$\mathbf{h}_{attn} = \mathbf{h} + \alpha \cdot \mathbf{A}, \quad \alpha \text{ is learnable}$$

A feed-forward network with GELU activation is then applied:

$$\mathbf{h}_{norm2} = \text{LayerNorm}(\mathbf{h}_{attn})$$

$$\mathbf{h}_{ff} = \text{FFN}(\mathbf{h}_{norm2})$$

$$\mathbf{h}_{final} = \mathbf{h}_{attn} + \mathbf{h}_{ff}$$

In this work, multi-head self-attention is employed to enable the model to focus on multiple aspects of each input dimension:

$$\text{MultiHead}(\mathbf{Q}, \mathbf{K}, \mathbf{V}) = \text{Concat}(\text{head}_1, \ldots, \text{head}_h)W^O$$

$$\text{head}_i = \text{Attention}(\mathbf{Q}W_i^Q, \mathbf{K}W_i^K, \mathbf{V}W_i^V)$$

$$\text{Attention}(\mathbf{Q}, \mathbf{K}, \mathbf{V}) = \text{softmax}\left(\frac{\mathbf{Q}\mathbf{K}^T}{\sqrt{d_k}}\right)\mathbf{V}$$

**Four-stream processing**

The proposed architecture processes four feature types through parallel attention streams:

$$\mathbf{h}_1 = \text{AttentionBlock}(\mathbf{x}_1) \quad \text{(Protein 1)}$$

$$\mathbf{h}_2 = \text{AttentionBlock}(\mathbf{x}_2) \quad \text{(Protein 2)}$$

$$\mathbf{h}_{\text{int}} = \text{AttentionBlock}(\mathbf{x}_{\text{int}}) \quad \text{(Interaction)}$$

$$\mathbf{h}_{\text{diff}} = \text{AttentionBlock}(\mathbf{x}_{\text{diff}}) \quad \text{(Difference)}$$

Each attention stream is allowed to focus on the most relevant features for its specific input type. This design facilitates specialized processing of individual protein properties, interactions, and differences, while maintaining a consistent architecture across all streams.

## Cross-attention fusion

To enable information exchange between streams, cross-attention mechanisms are implemented as follows:
Protein Cross-Attention:

$$\mathbf{h}_1^{\text{cross}} = \text{CrossAttention}(\mathbf{h}_1, \mathbf{h}_2, \mathbf{h}_2)$$

$$\mathbf{h}_2^{\text{cross}} = \text{CrossAttention}(\mathbf{h}_2, \mathbf{h}_1, \mathbf{h}_1)$$

Feature Cross-Attention:

$$\mathbf{h}_{\text{int}}^{\text{cross}} = \text{CrossAttention}(\mathbf{h}_{\text{int}}, \mathbf{h}_{\text{diff}}, \mathbf{h}_{\text{diff}})$$

$$\mathbf{h}_{\text{diff}}^{\text{cross}} = \text{CrossAttention}(\mathbf{h}_{\text{diff}}, \mathbf{h}_{\text{int}}, \mathbf{h}_{\text{int}})$$

Residual connections are applied to the final stream representations to preserve the original information:

$$\mathbf{h}_i^{\text{final}} = \mathbf{h}_i + \mathbf{h}_i^{\text{cross}}$$

## Quad-stream fusion

After the final representations of each stream are computed, they are concatenated to form a combined feature matrix:

$$\mathbf{h}_{\text{concat}} = \begin{bmatrix} \mathbf{h}_1^{\text{final}} \\ \mathbf{h}_2^{\text{final}} \\ \mathbf{h}_{\text{int}}^{\text{final}} \\ \mathbf{h}_{\text{diff}}^{\text{final}} \end{bmatrix} \in \mathbb{R}^{4 \times \text{hidden\_dim}} \tag{6}$$

The concatenated representation is then passed through a two-layer fusion network with GELU activation, which reduces dimensionality while preserving critical information:

$$\mathbf{h}_{\text{fused}} = \text{GELU}\Big( W_2 \cdot \text{GELU}(W_1 \mathbf{h}_{\text{concat}} + \mathbf{b}_1) + \mathbf{b}_2 \Big) \tag{7}$$

where the weight matrices and their dimensions are explicitly defined as:

$$W_1 \in \mathbb{R}^{2 \times \text{hidden\_dim} \times 4 \times \text{hidden\_dim}}$$

$$W_2 \in \mathbb{R}^{\text{hidden\_dim} \times 2 \times \text{hidden\_dim}}$$

and $\mathbf{b}_1$, $\mathbf{b}_2$ are the corresponding bias vectors.

This fusion network integrates information from all four streams Protein 1, Protein 2, Interaction, and Difference into a single, compact representation while retaining the most important features for downstream prediction tasks.

**Multi-task learning framework**

A multi-task learning framework is employed to simultaneously predict multiple aspects of protein–protein interactions from the fused representation $\mathbf{h}_{\text{fused}}$. Predictions for four related tasks are generated from the fused representation: interaction probability, uncertainty, binding strength, and interaction type.

The interaction probability is predicted using a sigmoid activation:

$$p_{\text{interact}} = \sigma(W_{\text{int}}\mathbf{h}_{\text{fused}} + \mathbf{b}_{\text{int}}), \tag{8}$$

and optimized via binary cross-entropy:

$$\mathcal{L}_{\text{interact}} = -\frac{1}{N} \sum_{i=1}^{N} \left[ y_i \log p_{\text{interact},i} + (1 - y_i) \log(1 - p_{\text{interact},i}) \right] . \tag{9}$$

The uncertainty head predicts a scalar $u_i$ per protein pair using a Softplus activation:

$$u_i = \text{Softplus}(W_{\text{unc}}\mathbf{h}_{\text{fused}} + \mathbf{b}_{\text{unc}}), \tag{10}$$

and the uncertainty loss is defined as:

$$\mathcal{L}_{\text{uncertainty}} = \frac{1}{N} \sum_{i=1}^{N} \begin{cases} u_i, & \text{if } \hat{y}_i = y_i \\ -u_i, & \text{otherwise} \end{cases} , \quad \hat{y}_i = \mathbb{1}\{p_{\text{interact},i} > 0.5\}. \tag{11}$$

This encourages the model to assign lower uncertainty to correct predictions and higher uncertainty to incorrect ones, improving calibration.

The binding strength target is computed directly from the predicted interaction probability and true interaction label, reflecting the confidence in interacting pairs:

$$s_{\text{target},i} = 2 \cdot \left| p_{\text{interact},i} - 0.5 \right| \cdot y_i, \tag{12}$$

while the predicted binding strength is

$$s_i = \sigma(W_{\text{bind}}\mathbf{h}_{\text{fused}} + \mathbf{b}_{\text{bind}}), \tag{13}$$

and optimized via mean squared error:

$$\mathcal{L}_{\text{binding}} = \frac{1}{N} \sum_{i=1}^{N} (s_i - s_{\text{target},i})^2 . \tag{14}$$

This ensures that binding strength is emphasized for pairs predicted with high interaction confidence, while non-interacting pairs are assigned zero, aligning with biologically meaningful interactions.

The interaction type is predicted using a Softmax activation:

$$\mathbf{t}_i = \text{Softmax}(W_{\text{type}}\mathbf{h}_{\text{fused}} + \mathbf{b}_{\text{type}}). \tag{15}$$

If the true interaction type is known, standard cross-entropy is used:

$$\mathcal{L}_{\text{type}} = -\frac{1}{N}\sum_{i=1}^{N}\sum_{c=1}^{C} y_{i,c}^{\text{type}} \log t_{i,c}. \tag{16}$$

If the type is unknown, the model is regularized toward a uniform distribution using KL-divergence:

$$\mathcal{L}_{\text{type}} = \text{KL}(\mathbf{t}_i \parallel \mathbf{u}), \quad \mathbf{u} = \frac{1}{C}\mathbf{1}_C. \tag{17}$$

Finally, the total multi-task loss combines all four components with tunable weights $(\lambda_1, \lambda_2, \lambda_3, \lambda_4)$:

$$\mathcal{L}_{\text{total}} = \lambda_1 \mathcal{L}_{\text{interact}} + \lambda_2 \mathcal{L}_{\text{uncertainty}} + \lambda_3 \mathcal{L}_{\text{binding}} + \lambda_4 \mathcal{L}_{\text{type}}. \tag{18}$$

This formulation directly mirrors the implementation in the code, ensuring that each task-specific computation and target is explicitly defined and justified with respect to its role in learning biologically meaningful protein–protein interactions.

## Optimization and hyperparameter search strategy

A systematic grid search integrated with advanced optimization techniques is employed to determine the optimal configuration of the ProtAttn-QuadNet architecture. The search space includes optimizer selection, learning rate, weight decay, batch size, learning rate scheduling, and multi-task loss weighting.

Four learning rates (0.0001, 0.0005, 0.001, 0.002), three weight decays ($1 \times 10^{-5}$, $1 \times 10^{-4}$, $1 \times 10^{-3}$), three optimizers (Adam, AdamW, RMSprop), three schedulers (ReduceLROnPlateau, CosineAnnealing, StepLR), and four batch sizes (32, 64, 128, 256) are explored. Additionally, three pre-configured settings are evaluated to assess stability and performance consistency.

- Conservative: LR = 0.0005, WD = 1e-4, AdamW, ReduceLROnPlateau

- Aggressive: LR = 0.001, WD = 1e-3, Adam, CosineAnnealing

- Balanced: LR = 0.0001, WD = 1e-5, RMSprop, StepLR

**Table 1. Optimal configurations for balanced and oversampled datasets.**

| Parameter | Balanced Dataset | Oversampled Dataset |
|---|---|---|
| Optimizer | AdamW | RMSprop |
| Learning Rate | 0.0005 | 0.0001 |
| Weight Decay | $1 \times 10^{-4}$ | $1 \times 10^{-5}$ |
| Scheduler | ReduceLROnPlateau | StepLR |
| Batch Size | 128 | 256 |

The best performing configuration is presented in Table 1.

## Results and discussions

### Evaluation metrics

Evaluating the performance of our protein–protein interaction (PPI) prediction model is essential for assessing its ability to identify true interactions, minimize false predictions, and generalize to unseen protein pairs. We employed five standard performance metrics: accuracy, precision, recall, F1-score, and AUC-ROC. These metrics collectively assess the model's overall correctness, reliability of positive predictions, sensitivity to true interactions, and balance between precision and recall. Performance was monitored across all training epochs to ensure stable learning and robust generalization.

### Performance evaluation of the proposed model

We evaluated the performance of the proposed protein–protein interaction (PPI) prediction model using two different datasets: a balanced dataset and an oversampled dataset. The balanced dataset allows us to assess how well the model performs when interacting and non-interacting protein pairs are equally represented, while the oversampled dataset increases the number of rare interactions, enabling us to evaluate how the model handles imbalanced data scenarios.

The comparative summary of model performance is presented in Table 2, highlighting the improvement gained through oversampling and confirming the proposed model's reliability for large-scale PPI prediction tasks.

On the balanced dataset, the model performed consistently on both validation and test sets, achieving high accuracy, precision, recall, F1-score, and AUC-ROC. These results indicate that the model effectively identifies true interactions while maintaining a low false-positive rate. The high AUC-ROC further demonstrates strong discriminative power, confirming that the model accurately differentiates between interacting and non-interacting protein pairs. The validation curves (Fig 2) illustrate stable learning across epochs.

When trained on the oversampled dataset, the model achieved even higher overall performance on both validation and test sets. The results indicate excellent accuracy, precision, recall, F1-score, and AUC-ROC, reflecting the model's strong ability to identify true interactions with minimal false positives. The accuracy progression shown in Fig 3 illustrates stable improvement throughout training. These findings suggest that the oversampling strategy effectively enhanced the model's capacity to learn from minority interaction samples without introducing overfitting.

The results demonstrate that the proposed model achieves highly competitive performance across both balanced and oversampled datasets. The oversampling strategy enhances recall and F1-score, improving detection of rare protein–protein interactions while maintaining strong precision. Consistently high AUC-ROC values further validate the model's robust classification ability, and the confusion matrices in Figs 4 and 5 show that most predictions fall along the true positive and true negative axes, indicating minimal misclassification. These findings establish that the model is effective on well-balanced data and remains reliable under imbalanced conditions, making it a promising framework for large-scale protein–protein interaction discovery and downstream bioinformatics applications.

Table 2.  Comparison of model performance on balanced and oversampled datasets.

| Dataset | Phase | Accuracy (%) | Precision (%) | Recall (%) | F1-score (%) | AUC-ROC (%) |
|---|---|---|---|---|---|---|
| Balanced | Validation | 97.18 | 96.36 | 98.01 | 97.18 | 99.10 |
| | Test | 97.16 | 96.51 | 97.81 | 97.16 | 99.00 |
| Oversampled | Validation | 99.23 | 98.62 | 99.84 | 99.22 | 99.80 |
| | Test | 99.19 | 98.66 | 99.70 | 99.18 | 99.76 |

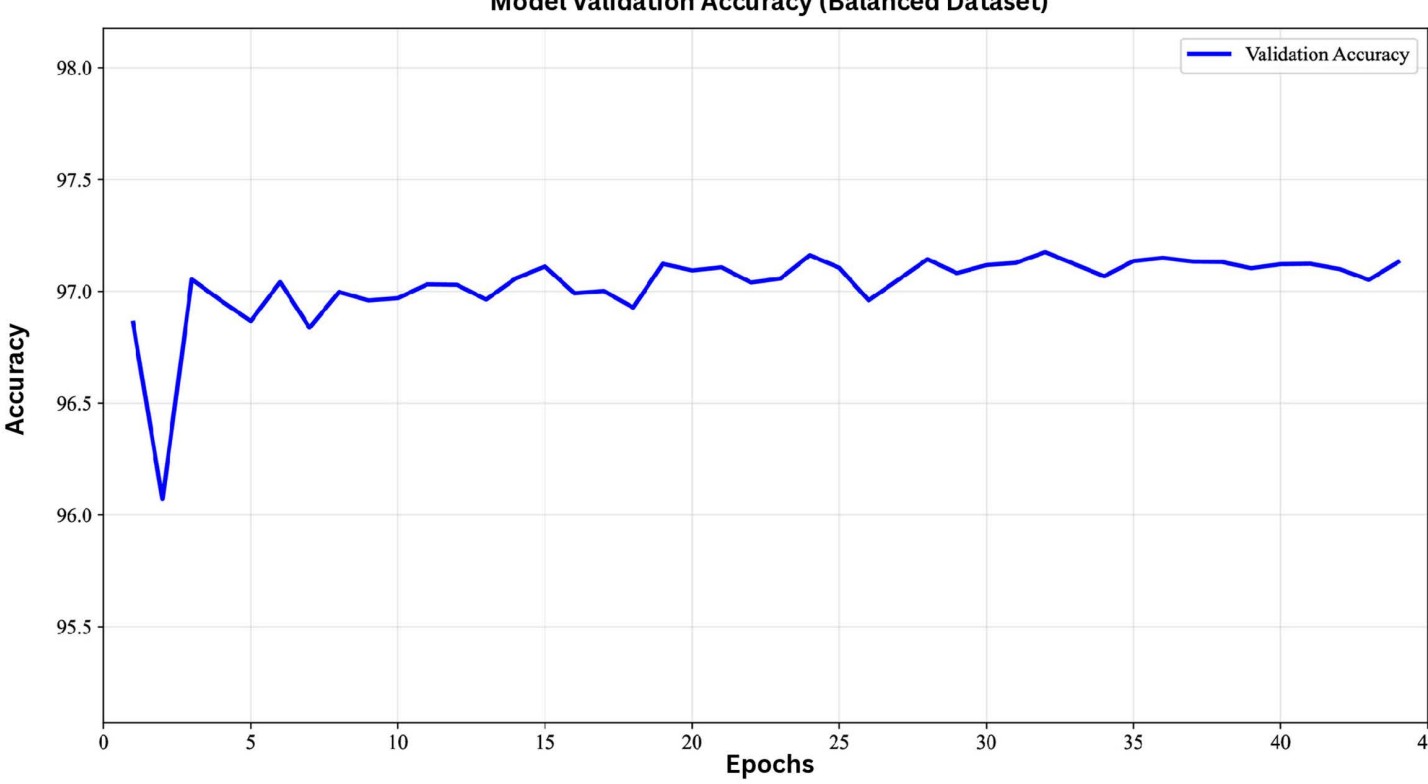

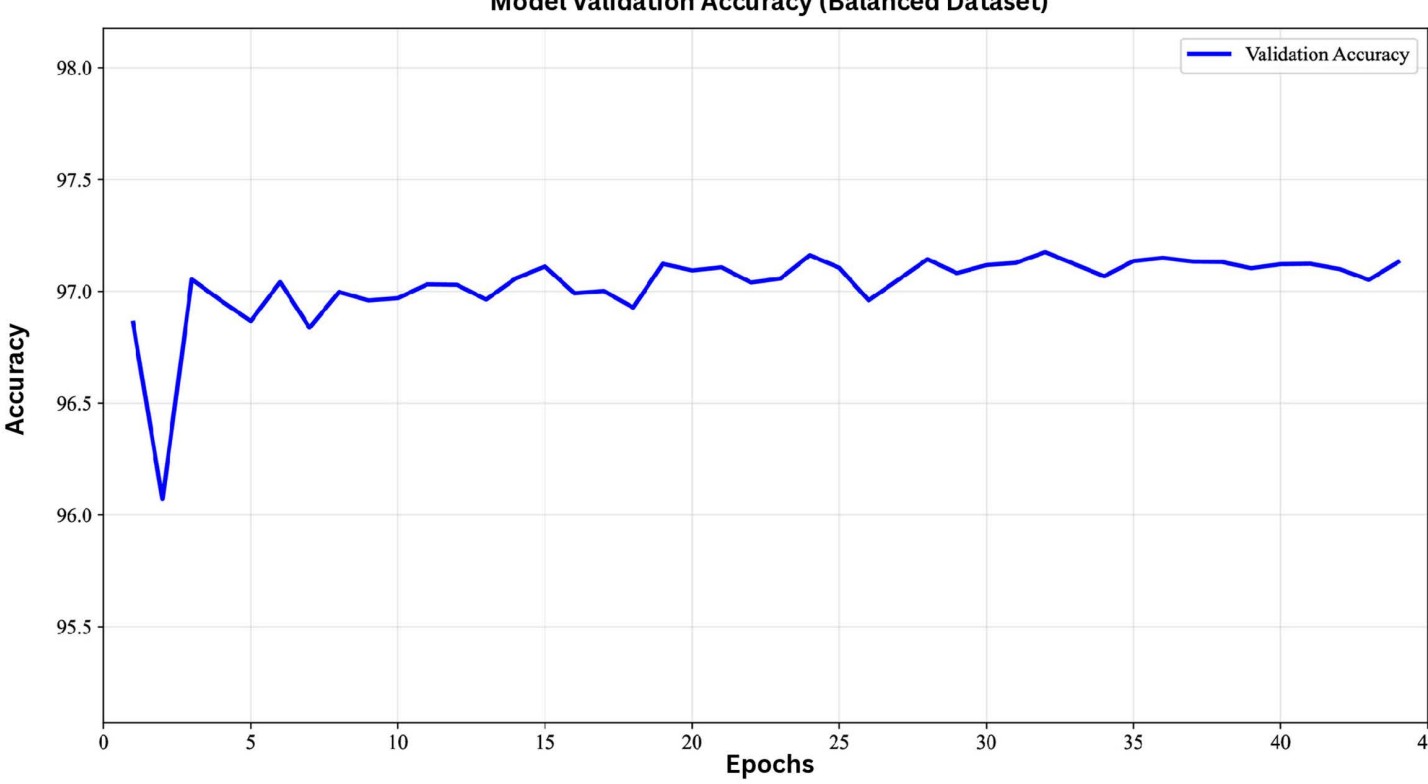

**Fig 2. Validation accuracy of the model using the balanced dataset.**

The performance of the proposed ProtAttn-QuadNet model was compared against several state-of-the-art protein-protein interaction prediction methods across multiple species and datasets (Table 3).

ProtAttn-QuadNet demonstrates strong predictive performance across multiple organisms. For the large-scale cross-species UniProt dataset, which contains over 541,000 positive and 541,000 negative interactions, ProtAttn-QuadNet achieved an accuracy of 99.19%, precision of 98.66%, recall of 99.70%, and an F1-score of 99.18%. While smaller organism-specific datasets reported in previous studies differ in size and composition, the organisms partially overlap with those included in our evaluation (e.g., *S. cerevisiae* and *H. pylori*). Our results indicate that ProtAttn-QuadNet achieves F1-scores that are comparable to or higher than the highest reported values in these organisms, demonstrating its robustness and generalization across diverse protein sequences.

## Statistical analysis

Comprehensive statistical analyses were performed to rigorously evaluate the reliability and significance of the proposed PPI prediction model. Three primary statistical tests were applied: the Chi-square test to assess statistical associations, the Wilcoxon signed-rank test to compare model performances, and an effect size analysis to determine the magnitude of observed differences. All statistical tests were conducted at a significance level of 0.05.

The Chi-square ($\chi^2$) test was applied to examine whether a statistically significant association existed between the predicted and true PPI classes (i.e., interacting vs. non-interacting protein pairs). The hypotheses were defined as:

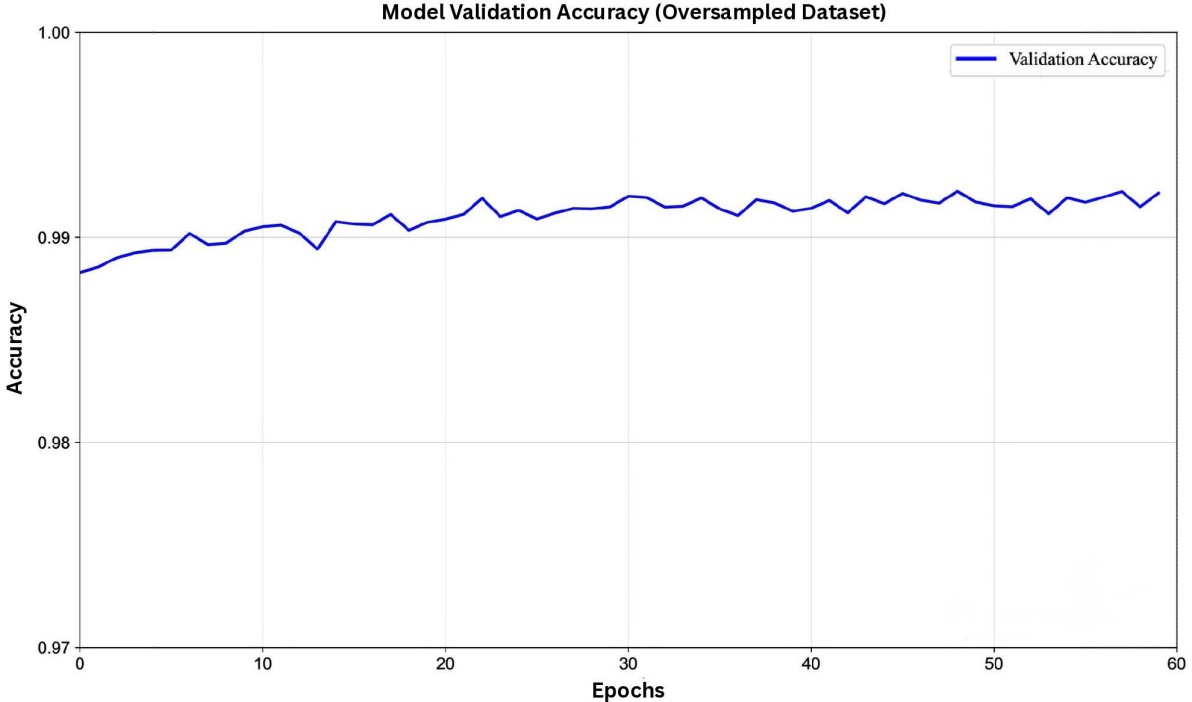

**Fig 3. Validation accuracy of the model using the oversampled dataset.**

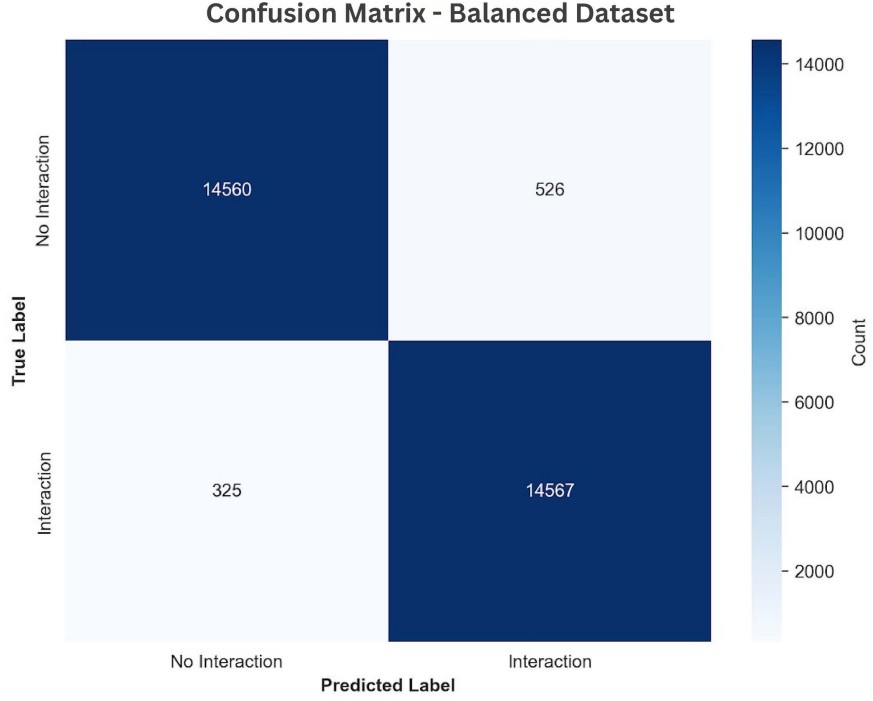

TN=14560, FP=526, FN=325, TP=14567

**Fig 4. Confusion matrix on the balanced dataset.**

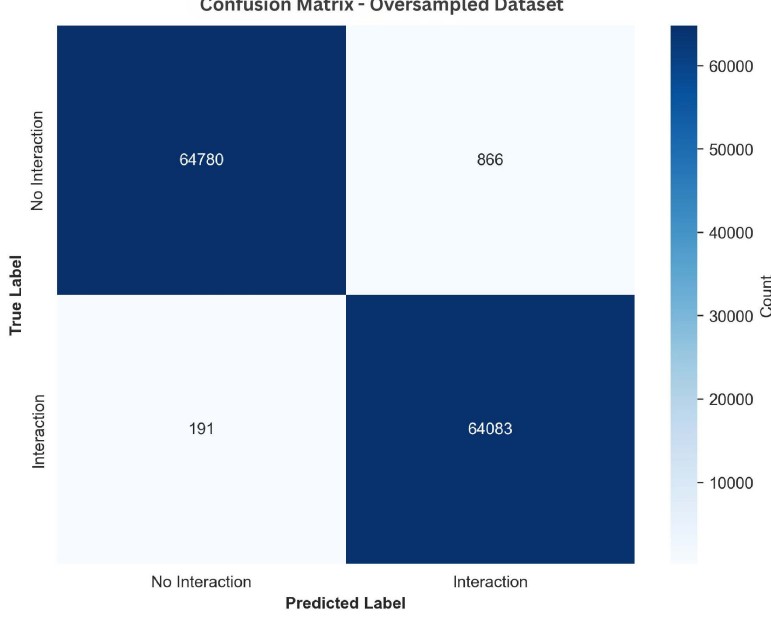

**Fig 5. Confusion matrix on the oversampled dataset.**

**Table 3. Comparison of ProtAttn-QuadNet model with existing state-of-the-art models.**

| Method | Dataset Details | ACC (%) | Precision (%) | Recall (%) | F1 (%) |
|---|---|---|---|---|---|
| DeepPPI (2017) [16] | *Saccharomyces cerevisiae* – 17,257 pos., 48,594 neg. | 92.50 | 94.38 | 90.56 | 92.52 |
| PIPR (2019) [18] | *S. cerevisiae* – 5,594 pos., 5,594 neg. | 97.09 | 97.00 | 97.17 | 97.09 |
| ADH-PPI (2022) [26] | *S. cerevisiae* – 5,594 pos., 5,594 neg. | 95.73 | 95.75 | 93.94 | 94.84 |
| | *Helicobacter pylori* – 1,458 pos., 1,365 neg. | 92.63 | 92.84 | 96.09 | 94.44 |
| SemiGNN-PPI (2023) [27] | *Homo sapiens (SHS27K)* – 12,517 total | – | – | – | 92.40 |
| | *H. sapiens (SHS148K)* – 44,488 total | – | – | – | 89.51 |
| xCAPT5 (2024) [22] | *H. pylori* – 1,458 pos., 1,365 neg. | 97.27 | 97.30 | 97.07 | 97.18 |
| | *S. cerevisiae* – 5,594 pos., 5,594 neg. | 99.76 | 99.76 | 99.75 | 99.37 |
| | *H. sapiens* – 27,593 pos., 34,298 neg. | 99.77 | 99.75 | 99.75 | 99.62 |
| SCMPPI (2024) [28] | *S. cerevisiae* – 11,188 total | 94.55 | 96.68 | 92.24 | 94.40 |
| HI-PPI (2025) [29] | *H. sapiens* (SHS27K) – 12,517 total | 85.82 | – | – | 79.57 |
| | *H. sapiens* (SHS148K) – 44,488 total | 89.32 | – | – | 84.12 |
| ProtAttn-QuadNet (Ours) | Cross-Species (*UniProt*) – 541,331 pos., 541,331 neg. | 99.19 | 98.66 | 99.70 | 99.18 |

$$H_0 : \text{Predicted and actual PPI classes are independent}$$

$$H_1 : \text{Predicted and actual PPI classes are dependent}$$

The Chi-square statistic was computed as:

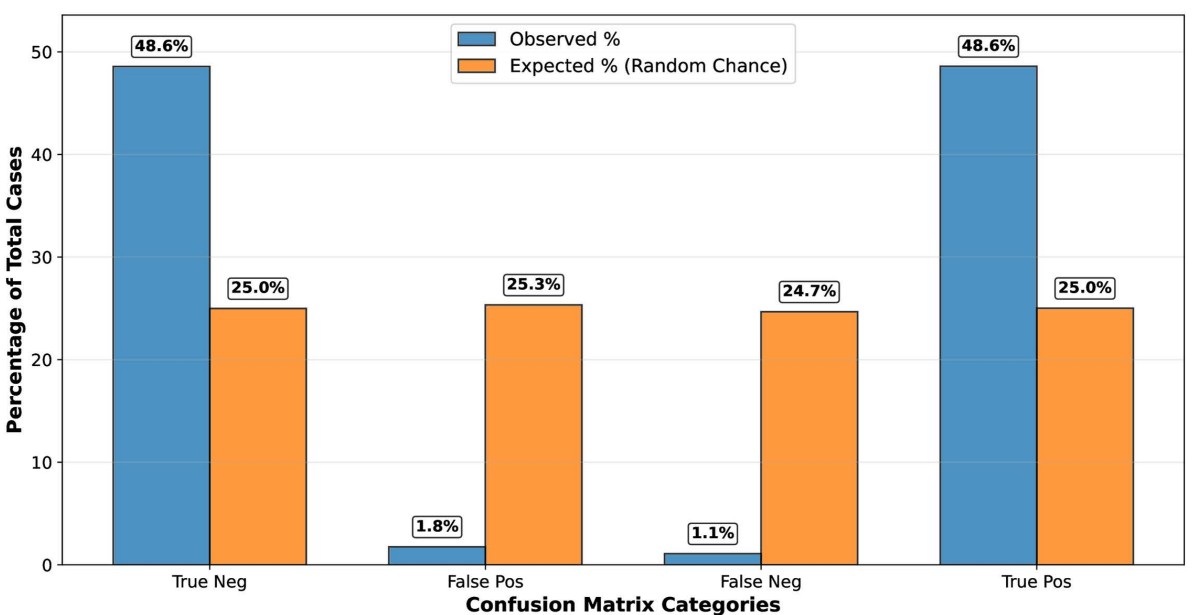

**Fig 6. Distribution of predicted versus actual protein–protein interaction categories.**

$$\chi^2 = \sum \frac{(O_i - E_i)^2}{E_i}$$

where $O_i$ and $E_i$ represent the observed and expected frequencies, respectively.

The analysis produced a $\chi^2$ value of 26,671.80 with 1 degree of freedom, $p < 0.000001$, and a Cramér's $V$ of 0.944, indicating a very strong statistical association between the predicted and actual PPI outcomes.

Fig 6 illustrates the percentage distribution across confusion matrix categories, clearly showing the deviation from random classification behavior.

To confirm that the proposed PPI model significantly outperformed the random-chance baseline of 50%, we applied a Wilcoxon signed-rank test on the distributions of accuracy, F1-score, and AUC-ROC. Each metric yielded a test statistic of $W = 990.0$ with $p = 5.68 \times 10^{-14}$, indicating that the model's performance gains were highly significant. Building on this, we quantified the strength of association and the magnitude of the predictive effect using multiple effect size measures. Cramér's $V$, Cohen's $w$, and the Phi coefficient were all 0.944, reflecting a *very large* effect size. These results collectively demonstrate that the proposed PPI model not only significantly outperforms a random baseline but also produces predictions that are strongly consistent with true interaction labels, highlighting its practical and statistical relevance.

## Author contributions

**Conceptualization:** Md. Shahidul Islam.

**Data curation:** Md. Raihan Kabir.

**Formal analysis:** Md. Shahidul Islam.

**Methodology:** Md. Muhtasim Rahman Mim, Md. Raihan Kabir.

**Resources:** Md. Shahidul Islam.

**Software:** Md. Muhtasim Rahman Mim.

**Supervision:** Md. Shahidul Islam.

**Validation:** Md. Shahidul Islam.

**Visualization:** Md. Raihan Kabir.

**Writing – original draft:** Md. Muhtasim Rahman Mim, Md. Raihan Kabir.

**Writing – review & editing:** Md. Shahidul Islam, Md. Muhtasim Rahman Mim, Md. Raihan Kabir.

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
