## [Decision Letter · Decision Letter 0]

26 Jan 2026

PONE-D-25-63548ProtAttn-QuadNet: An attention-based deep learning framework for protein–protein interaction prediction using ProtBERT embeddingsPLOS One

Dear Dr. Islam,

Thank you for submitting your manuscript to PLOS ONE. After careful consideration, we feel that it has merit but does not fully meet PLOS ONE’s publication criteria as it currently stands. Therefore, we invite you to submit a revised version of the manuscript that addresses the points raised during the review process.

We look forward to receiving your revised manuscript.

Kind regards,

Musa Aydin, Ph.D.

Academic Editor

PLOS One

Journal Requirements:

Reviewers' comments:

Reviewer's Responses to Questions

**Comments to the Author**

1. Is the manuscript technically sound, and do the data support the conclusions?

Reviewer #1: Yes

Reviewer #2: Partly

2. Has the statistical analysis been performed appropriately and rigorously? 

Reviewer #1: Yes

Reviewer #2: Yes

3. Have the authors made all data underlying the findings in their manuscript fully available?

Reviewer #1: Yes

Reviewer #2: Yes

4. Is the manuscript presented in an intelligible fashion and written in standard English?

Reviewer #1: Yes

Reviewer #2: Yes

5. Review Comments to the Author

Reviewer #1: Page 7, Abstract: Align accuracy and AUC values with those in the Results section.

Page 10, Table 1: Consider adding standard deviations if multiple runs were performed.

Page 13, Data Preprocessing: Clarify why robust scaling was chosen over other methods (e.g., MinMax, StandardScaler) in more detail.

Page 14, Feature Engineering: Briefly justify the noise injection (Eq. 8) and its impact on generalization.

References: Ensure all in-text citations are in the reference list and formatted consistently.

Reviewer #2: • The model presents an innovative approach to protein representation for PPI tasks, utilizing complementary methods to capture both similarity and difference information between protein pairs.

• The extracted representations are processed through attention mechanisms in multiple configurations, then trained within a multi-task learning framework that derives information from binary interaction labels.

• Figures are not properly integrated into the text and instead appear individually on separate pages following the references.

• Section ordering should follow standard convention: Results and Discussion must appear after Materials and Methods.

• While evaluation metrics can be briefly mentioned as standard, their definitions need not be elaborated.

• Table 1 values should not be redundantly listed in the text. Commentary on the achieved metrics is appropriate, but verbatim repetition of tabular data is unnecessary.

• There is a critical problem with comparison to other models. It is stated in the article verbatim “Across all evaluated datasets, ProtAttn-QuadNet demonstrated consistently superior predictive performance”. This statement is not proven at all. Only given result for ProtAttn-QuadNet model is on UniProt dataset. But no other model is evaluated on this dataset, making a direct comparison impossible. It is also stated “Notably, even on smaller organism-specific datasets, such as S. cerevisiae and H. pylori, ProtAttn-QuadNet either matched or exceeded the highest reported F1-scores from previous studies, including xCAPT5 and SCMPPI.”. Results achieved by ProtAttn-QuadNet in these datasets are not shared anywhere in the article. Authors must share these results in order to support this claim.

• Figure 6 contains multiple discrepancies with both the textual description and implementation. Minor issues include omission of projection layers and inconsistent naming (only x_diff transforms to h_diff while other inputs retain their names). More substantially, the attention mechanism connections are unclear—all inputs appear to interact directly between the attention and hidden layers, which contradicts both the text and code. The hidden layer following the attention layer is ambiguous: if this represents the FFN applied after MultiHeadAttention, why are these components not grouped into an AttentionBlock as described in the text? Given its central importance, this figure requires careful revision.

• Notation for A is used inconsistently. Initially, A denotes the output of MultiHeadAttention used in computing h. Subsequently, A represents the second output of AttentionBlock (the attention weights in the code, which remain unused). This creates ambiguity suggesting the MultiHeadAttention output is split for separate use. These should be labeled differently. Furthermore, the purpose of extracting attention weights is unclear given they are utilized neither in the code nor discussed in the manuscript.

• Tasks beyond interaction prediction require more detailed specification, particularly their exact computational formulation. For example, binding strength is computed in the code as: interaction_conf = torch.abs(predictions['interaction'].squeeze() - 0.5) * 2; binding_target = interaction_conf * targets.float(). Such calculations should be explicitly presented and justified in the methodology.

• Axis labels and plot titles should be consistent between figures 1 and 2.

• Bias notation alternates between bold and italic; one convention should be used throughout.

6. PLOS authors have the option to publish the peer review history of their article (what does this mean?). If published, this will include your full peer review and any attached files.

Reviewer #1: **Yes:** Ghassan Abdul-Majeed

Reviewer #2: No

---

## [Author Response · Author response to Decision Letter 1]

24 Mar 2026

Reviewer #1

Thank you for your careful review and valuable feedback. We sincerely appreciate your time and insightful suggestions. We have addressed all your comments and revised the manuscript accordingly.

Please note that the manuscript has been restructured to comply with the journal guidelines; consequently, the page and equation numbering may differ in the revised version.

Page 7, Abstract: Align accuracy and AUC values with those in the Results section.

Response: Thank you very much for your thoughtful comments. We have carefully rechecked the reported values and confirm that they are correct for the independent test set. However, to avoid any potential confusion between the test and validation sets, we have revised the corresponding text in the manuscript as follows:

“Evaluated on the independent test set of a large-scale dataset from UniProt, ProtAttn-QuadNet achieves 97.16% accuracy (AUC-ROC 99.00%) on balanced data and 99.19% accuracy (AUC-ROC 99.76%) on oversampled datasets, surpassing several recent state-of-the-art PPI prediction methods. ”

Page 10, Table 1: Consider adding standard deviations if multiple runs were performed.

Response: Thank you for your comment. We did not perform a statistical deviation test; however, we will consider including it in the final version.

Page 13, Data Preprocessing: Clarify why robust scaling was chosen over other methods (e.g., MinMax, StandardScaler) in more detail.

Response: Thank you for the helpful suggestion. We have expanded the explanation in the manuscript to clarify why RobustScaler was selected over StandardScaler and MinMaxScaler, emphasizing that protein embeddings may exhibit non-Gaussian distributions and outliers, making IQR–based scaling more stable for such features. Please see the highlighted verison of the manuscript (page 3, Data Processing).

Page 14, Feature Engineering: Briefly justify the noise injection (Eq. 8) and its impact on generalization.

Response: We have added a new paragraph titled “Gaussian Noise Injection for Data Augmentation” before Eq. 3 (previously Eq. 8) to clarify the rationale for noise augmentation. The paragraph explains that small Gaussian perturbations to ProtBert embeddings improve model robustness and generalization by encouraging the model to learn stable interaction patterns rather than memorizing exact embedding values. Please see page 4 in the manuscript.

References: Ensure all in-text citations are in the reference list and formatted consistently.

Response: Thank you for your comment. We have carefully reviewed the reference list and confirm that all references are properly cited within the text and are formatted appropriately.

Reviewer #2

We sincerely thank you for your thorough review and insightful feedback. We greatly appreciate the time and effort you invested in evaluating our manuscript. We have carefully addressed all your comments and have updated the manuscript accordingly.

• The model presents an innovative approach to protein representation for PPI tasks, utilizing complementary methods to capture both similarity and difference information between protein pairs.

Response: Thank you very much for your comment.

• The extracted representations are processed through attention mechanisms in multiple configurations, then trained within a multi-task learning framework that derives information from binary interaction labels.

Response: Thank you. We appreciate your feedback.

• Figures are not properly integrated into the text and instead appear individually on separate pages following the references.

Response: Thank you very much for your valuable comment. However, we have followed the PLOS ONE submission guidelines, which state: “Do not include figures in the main manuscript file. Each figure must be prepared and submitted as an individual file.” Therefore, we have prepared and submitted the figures separately in accordance with the journal’s requirements.

• Section ordering should follow standard convention: Results and Discussion must appear after Materials and Methods.

Response: Thank you very much for your careful observation. We have revised the manuscript to comply with the PLOS ONE submission guidelines. The Materials and Methods section has now been placed before the Results and Discussion section as required.

• While evaluation metrics can be briefly mentioned as standard, their definitions need not be elaborated.

Response: As suggested, we have removed the detailed formulas and now briefly describe the evaluation metrics.

• Table 1 values should not be redundantly listed in the text. Commentary on the achieved metrics is appropriate, but verbatim repetition of tabular data is unnecessary.

Response: Thank you for your valuable comment. We have revised the text to avoid verbatim repetition of the values in Table 1. Instead, we now provide a concise commentary on the model’s performance, highlighting trends and key observations while referring to the table for detailed metrics. Please see page 9 in the manuscript.

• There is a critical problem with comparison to other models. It is stated in the article verbatim “Across all evaluated datasets, ProtAttn-QuadNet demonstrated consistently superior predictive performance”. This statement is not proven at all. Only given result for ProtAttn-QuadNet model is on UniProt dataset. But no other model is evaluated on this dataset, making a direct comparison impossible. It is also stated “Notably, even on smaller organism-specific datasets, such as S. cerevisiae and H. pylori, ProtAttn-QuadNet either matched or exceeded the highest reported F1-scores from previous studies, including xCAPT5 and SCMPPI.”. Results achieved by ProtAttn-QuadNet in these datasets are not shared anywhere in the article. Authors must share these results in order to support this claim.

Response: Thank you so much. We have clarified in the revised manuscript that comparisons with previous studies are based on overlapping organisms rather than identical datasets. Please see the revised text in the manuscript.

• Figure 6 contains multiple discrepancies with both the textual description and implementation. Minor issues include omission of projection layers and inconsistent naming (only x_diff transforms to h_diff while other inputs retain their names). More substantially, the attention mechanism connections are unclear—all inputs appear to interact directly between the attention and hidden layers, which contradicts both the text and code. The hidden layer following the attention layer is ambiguous: if this represents the FFN applied after MultiHeadAttention, why are these components not grouped into an AttentionBlock as described in the text? Given its central importance, this figure requires careful revision.

Response: Thank you so much for your critical observation. We have now updated the figure according to the text to remove any confusion. Which also removed the connection issues.

• Notation for A is used inconsistently. Initially, A denotes the output of MultiHeadAttention used in computing h. Subsequently, A represents the second output of AttentionBlock (the attention weights in the code, which remain unused). This creates ambiguity suggesting the MultiHeadAttention output is split for separate use. These should be labeled differently. Furthermore, the purpose of extracting attention weights is unclear given they are utilized neither in the code nor discussed in the manuscript.

Response: We sincerely thank you for carefully identifying this issue, which has helped us improve the consistency of the manuscript.

Each AttentionBlock returns the transformed feature representation (h) and the corresponding attention weights (A). We used only the feature representations (h) for downstream prediction. The attention weights are present in the implementation for potential analysis; however, they are neither captured during the function call nor utilized in subsequent computations. Therefore, references to the attention weights have been removed from the manuscript, and the notation A now consistently refers only to the output of MultiHeadAttention.

• Tasks beyond interaction prediction require more detailed specification, particularly their exact computational formulation. For example, binding strength is computed in the code as: interaction_conf = torch.abs(predictions['interaction'].squeeze() - 0.5) * 2; binding_target = interaction_conf * targets.float(). Such calculations should be explicitly presented and justified in the methodology.

Response: Thank you for your comment. We have now discussed the loss function in details including its exact formulation. Please see equations 13 to 18 in the manuscript (page 7).

• Axis labels and plot titles should be consistent between figures 1 and 2.

Response: Thank you for the comment. We have now made both figures 1 and 2 consistent in title and axis labels.

• Bias notation alternates between bold and italic; one convention should be used throughout.

Response: Thank you for your careful observation. We have revised the manuscript to ensure consistent notation throughout. Specifically, all bias terms are now presented in bold font, as they represent vectors.

---

## [Decision Letter · Decision Letter 1]

1 May 2026

ProtAttn-QuadNet: An attention-based deep learning framework for protein–protein interaction prediction using ProtBERT embeddings

PONE-D-25-63548R1

Dear Dr. Islam,

We’re pleased to inform you that your manuscript has been judged scientifically suitable for publication and will be formally accepted for publication once it meets all outstanding technical requirements.

Kind regards,

Musa Aydin, Ph.D.

Academic Editor

PLOS One

Additional Editor Comments (optional):

Reviewers' comments:

Reviewer's Responses to Questions

**Comments to the Author**

1. If the authors have adequately addressed your comments raised in a previous round of review and you feel that this manuscript is now acceptable for publication, you may indicate that here to bypass the “Comments to the Author” section, enter your conflict of interest statement in the “Confidential to Editor” section, and submit your "Accept" recommendation.

Reviewer #1: All comments have been addressed

Reviewer #2: All comments have been addressed

2. Is the manuscript technically sound, and do the data support the conclusions?

Reviewer #1: Yes

Reviewer #2: Yes

3. Has the statistical analysis been performed appropriately and rigorously? 

Reviewer #1: Yes

Reviewer #2: Yes

4. Have the authors made all data underlying the findings in their manuscript fully available?

Reviewer #1: Yes

Reviewer #2: Yes

5. Is the manuscript presented in an intelligible fashion and written in standard English?

Reviewer #1: Yes

Reviewer #2: Yes

6. Review Comments to the Author

Reviewer #1: Dear Authors,

Thank you for fixing all my comments. The revised manuscript is now much better than the first one.

Reviewer #2: (No Response)

7. PLOS authors have the option to publish the peer review history of their article (what does this mean?). If published, this will include your full peer review and any attached files.

Reviewer #1: **Yes:** Ghassan Abdul-Majeed

Reviewer #2: No

---

## [Editor Report · Acceptance letter]

PONE-D-25-63548R1

PLOS One

Dear Dr. Islam,

I'm pleased to inform you that your manuscript has been deemed suitable for publication in PLOS One. Congratulations! Your manuscript is now being handed over to our production team.

Kind regards,

on behalf of

Assoc. Prof. Musa Aydin

Academic Editor

PLOS One